Pre- versus post-exercise protein intake has similar effects on muscular adaptations

Schoenfeld Brad Jon highnrg123@aol.com bradschoenfeldphd@gmail.com 1
Aragon Alan 2
Wilborn Colin 3
Urbina Stacie L. 3
Hayward Sara E. 3
Krieger James 4
1 Department of Health Sciences, Herbert H. Lehman College, City University of New York , Bronx , NY , United States
2 Department of Nutrition, California State University , Northridge , CA , United States
3 Graduate School and Research, University of Mary Hardin Baylor , Belton , TX , United States
4 Weightology , Issaquah , WA , United States
Keogh Justin
Electronic publication date: 2017 Jan 3
Publication date: 2017
Volume: 5
Electronic Location ID: e2825
Received 2016 Sep 12; Accepted 2016 Nov 22
Copyright: ©2017 Schoenfeld et al.
Copyright year: 2017
Copyright holder: Schoenfeld et al.
License: This is an open access article distributed under the terms of the Creative Commons Attribution License, which permits unrestricted use, distribution, reproduction and adaptation in any medium and for any purpose provided that it is properly attributed. For attribution, the original author(s), title, publication source (PeerJ) and either DOI or URL of the article must be cited.
License URL: https://creativecommons.org/licenses/by/4.0/

Erratum in: Correction: Pre- versus post-exercise protein intake has similar effects on muscular adaptations 5 1 8 2017 e2825/correction-1 PeerJ PMC5539510 28785512
Keywords: Nutrient timing, Anabolic window, Resistance training, Protein timing, Protein supplementation

Funding: The authors received no funding for this work.

==============================
The purpose of this study was to test the anabolic window theory by investigating muscle strength, hypertrophy, and body composition changes in response to an equal dose of protein consumed either immediately pre- versus post-resistance training (RT) in trained men. Subjects were 21 resistance-trained men (>1 year RT experience) recruited from a university population. After baseline testing, participants were randomly assigned to 1 of 2 experimental groups: a group that consumed a supplement containing 25 g protein and 1 g carbohydrate immediately prior to exercise (PRE-SUPP) (n = 9) or a group that consumed the same supplement immediately post-exercise (POST-SUPP) (n = 12). The RT protocol consisted of three weekly sessions performed on non-consecutive days for 10 weeks. A total-body routine was employed with three sets of 8–12 repetitions for each exercise. Results showed that pre- and post-workout protein consumption had similar effects on all measures studied (p > 0.05). These findings refute the contention of a narrow post-exercise anabolic window to maximize the muscular response and instead lends support to the theory that the interval for protein intake may be as wide as several hours or perhaps more after a training bout depending on when the pre-workout meal was consumed.

Introduction

Nutrient timing, operationally defined as the consumption of nutrients in and/or around an exercise bout, has been advocated as a strategy to optimize a myriad of performance- and muscular-related adaptations. Several researchers have put forth the notion that the timing of nutrient consumption is even more important to these adaptations than the quantity of food and macronutrient ratio of the diet (Candow & Chilibeck, 2008). Perhaps the most heralded aspect of nutrient timing involves consuming protein immediately after exercise. The purported beneficial effects (i.e., increased muscle protein synthetic response) of protein timing are based on the hypothesis that a limited “anabolic window of opportunity” exists for post-workout anabolism (Lemon, Berardi & Noreen, 2002). To take advantage of this window of opportunity, common thought is that protein must be consumed within approximately 45 min to 1 h of completion of exercise to maximize post-workout muscle protein synthesis (MPS) (Ivy & Ferguson-Stegall, 2013). It has been postulated that the anabolic response to a resistance training bout is blunted if protein is ingested after this narrow window, thereby impairing muscular gains (Ivy & Ferguson-Stegall, 2013).

A review of literature determined that while compelling evidence exists showing muscle is sensitized to protein ingestion following a workout, the anabolic window does not appear to be as narrow as what was once thought (Aragon & Schoenfeld, 2013). Rather, the authors proposed that the interval for consumption may be as wide as 5–6 h after exercise depending on the timing of the pre-workout meal; the closer a meal is consumed prior to exercise, the larger the post-workout anabolic window of opportunity.

Research examining the existence of a narrow post-workout window is equivocal. In a study of healthy young and middle-aged subjects, Levenhagen et al. (2001) reported that protein synthesis of the legs and whole body, as determined by dilution and enrichment of phenylalanine, was increased threefold when an oral supplement containing 10 g protein, 8 g carbohydrate and 3 g fat was consumed immediately following exercise compared to just a 12% increase when the supplement was ingested 3-hours post-workout. It should be noted that the training protocol involved moderate intensity, long duration aerobic exercise, raising the possibility that results reflected mitochondrial and/or sarcoplasmic protein fractions, as opposed to synthesis of contractile elements (Kumar et al., 2009). Conversely, Rasmussen et al. (2000) found no significant difference in leg net amino acid balance when 6 g essential amino acids (EAA) were co-ingested with 35 g carbohydrate either 1 h or 3 h after resistance training. Given that the training protocol involved 18 sets of lower body resistance exercise, it can be inferred that findings were indicative of myofibrillar protein synthesis (Donges et al., 2012). Moreover, the amount of EAA was markedly higher in Rasmussen et al. versus Levenhagen et al., potentially confounding results between studies. It should be noted that while these studies provide an interesting snapshot of the transient post-exercise responses to protein timing, there is evidence that acute measures of MPS do not necessarily correlate with long-term increases in muscle growth (Adams & Bamman, 2012).

Longitudinal studies on the topic of protein timing are conflicting. A number of studies have shown beneficial effects of post-workout protein timing on muscle strength and size (Esmarck et al., 2001; Cribb & Hayes, 2006; Willoughby, Stout & Wilborn, 2007) while others have not (Hoffman et al., 2009; Candow et al., 2006; Verdijk et al., 2009). A recent meta-analysis by Schoenfeld, Aragon & Krieger (2013) found that consuming protein within 1 h post-resistance exercise had a small but significant effect on increasing muscle hypertrophy compared to delaying consumption by at least 2 h. However, sub-analysis of these results revealed the effect all but disappeared after controlling for the total intake of protein, indicating that favorable effects were due to unequal protein intake between the experimental and control groups (∼1.7 g/kg versus 1.3 g/kg, respectively) as opposed to temporal aspects of feeding. The authors noted that inherent limitations of the studies obscure the ability to draw definitive, evidence-based conclusions on the efficacy of protein timing. Specifically, only three studies in the meta-analysis met inclusion criteria for matched protein intake between experimental and control groups. Of these studies, one showed a significant benefit to protein timing while two showed no differences between groups. Compounding matters, only two of the matched studies investigated the effects of protein timing on well-trained subjects. Cribb & Hayes (2006) randomized a cohort of young recreational male bodybuilders to consume 1 g/kg of a supplement containing 40 g whey isolate, 43 g glucose, and 7 g creatine monohydrate either immediately before and after exercise versus in the early morning and late evening in young recreational male bodybuilders. After 10 weeks of progressive resistance exercise, significant increases in lean body mass and hypertrophy of type II fibers were seen when the supplement was timed around the exercise bout as compared to delaying consumption. On the other hand, Hoffman et al. (2009) showed no significant differences in total body mass or lean body mass when resistance-trained men with an average of 5.9 years lifting experience consumed a supplement containing 42 g protein and 2 g carbohydrate immediately before and after resistance exercise versus in the early morning and late evening over a 10-week period.

Therefore, the purpose of this study was to investigate muscular adaptations in response to an equal dose of protein consumed either immediately pre- versus post-resistance exercise in well-trained men. It was hypothesized that consuming protein prior to resistance training would negate the need to consume protein immediately post-workout for maximizing muscular adaptations.

Methods

Experimental approach to the problem

To determine the effects of pre- versus post-exercise protein consumption on muscular adaptations, resistance trained subjects were pair-matched according to baseline strength in the squat and bench press exercises and then randomly assigned to one of two experimental groups: a group that consumed a supplement containing 25 g protein and 1 g carbohydrate immediately prior to exercise (PRE-SUPP) or immediately after the exercise bout (POST-SUPP). Subjects in the PRE-SUPP group were instructed to refrain from eating for at least 3 h after the exercise bout while those in the POST-SUPP group were instructed to refrain from eating for at least 3 h prior to the exercise bout. All subjects performed a hypertrophy-type resistance training protocol consisting of three weekly sessions carried out on non-consecutive days for 10 weeks. A total-body routine was employed with three sets of 8–12 repetitions performed for each exercise. Subjects were tested prior to the initial training session (T1), at the mid-point of the study (T2), and after the final training session (T3) for measures of body composition, muscle thickness, and maximal strength.

Participants

Twenty-one male volunteers were recruited from a university population (age = 22.9 ± 3.0 years; height = 175.5 ± 5.9 cms; body mass = 82.9 ± 13.6 kgs). Subjects had no existing musculoskeletal disorders, were self-reported to be free from the use of anabolic steroids or any other illegal agents known to increase muscle size for the previous year, and were considered experienced lifters, defined as consistently lifting weights at least three times per week for a minimum of one year and regularly performing the bench press and squat exercises. Approval for the study was obtained from the University of Mary Hardin-Baylor Institutional Review Board (IRB). Informed consent was obtained from all participants.

Supplementation procedures

After baseline testing, participants were pair-matched according to baseline strength in the squat and bench press exercises and then randomly assigned to one of two experimental groups: a group that consumed a supplement containing 25 g protein and 1g carbohydrate (Iso100 Hydrolyzed Whey Protein Isolate, Dymatize Nutrition, Dallas, TX) immediately prior to exercise (PRE-SUPP) (n = 9) or immediately after the exercise bout (POST-SUPP) (n = 12). The chosen supplement was based on research showing that consumption of 20–25 g of whey protein maximizes the MPS response in young resistance trained men (Atherton & Smith, 2012; Breen & Phillips, 2012). All subjects consumed the supplement in the presence of a research assistant to ensure compliance. Subjects in the PRE-SUPP group were instructed to refrain from eating for at least 3 h after the exercise bout to ensure that consumption of a post-workout meal did not confound results. Similarly, those in the POST-SUPP group were instructed to refrain from eating for at least 3 h prior to the exercise bout to ensure that consumption of a pre-workout meal did not confound results.

Resistance training procedures

The resistance training protocol consisted of nine exercises per session. These exercises targeted the anterior torso muscles (flat barbell bench press, barbell military press), the posterior muscles of the torso (wide grip lat pulldown, seated cable row), the thigh musculature (barbell back squat, machine leg press, and machine leg extension), and upper extremities (dumbbell biceps curl, triceps pushdown). Subjects were instructed to refrain from performing any additional resistance-type training and to avoid additional aerobic-type exercise other than what was part of normal daily activities for the 10-week study period.

Training consisted of three weekly sessions performed on non-consecutive days for 10 weeks. All routines were directly supervised by research staff trained to ensure proper performance of all exercises. Intensity of load was approximately 75% of 1 repetition maximum (RM)—generally considered to equate to a 10RM (Baechle & Earle, 2008)—so that a target repetition range of 8–12 repetitions is achieved on each set. Prior to training, participants underwent 10RM testing to determine individual initial loads for each exercise. Repetition maximum testing was consistent with recognized guidelines as established by the National Strength and Conditioning Association (Baechle & Earle, 2008). Subjects performed three sets of each exercise. Sets were carried out to the point of momentary concentric muscular failure—the inability to perform another concentric repetition while maintaining proper form. Cadence of repetitions was carried out with a controlled concentric contraction and an approximately 2 s eccentric contraction as determined by the supervising member of the research team. Subjects were afforded 90 s rest between sets. The load was adjusted for each exercise as needed on successive sets to ensure that subjects achieved failure in the target repetition range. Attempts were made to progressively increase the loads lifted each week within the confines of maintaining the target repetition range.

Dietary intervention

To help ensure a maximal anabolic response, each subject was given a dietary plan (protein equating to 1.8 g/kg of body mass, fat equating to 25–30% of total energy intake, and the remaining calories in carbohydrate) designed to create an energy surplus of 500 kcal/day. Dietary adherence was assessed by self-reported food records using MyFitnessPal.com (http://www.myfitnesspal.com), which were collected and analyzed during each week of the study. Subjects were instructed on how to properly complete the logs and record all food items and their respective portion sizes that were consumed for the designated period of interest. Each item of food was individually entered into the program, and the program provided relevant information as to total energy consumption, as well as amount of energy derived from proteins, fats, and carbohydrates over the length of the study. Diet logs were recorded every day during the study. When calculating total calories, protein, carbohydrate, and fat, values were derived from the three days prior to each testing session (T1, T2, T3) and averaged. Subjects received ongoing counseling from the research staff at each session on the importance of maintaining the prescribed dietary regimen.

Measurements

Testing was conducted prior to the initial training session (T1), at the mid-point of the study (T2), and after the final training session (T3). Subjects were instructed to refrain from any strenuous exercise for at least 48 h prior to each testing session. Subjects were instructed to avoid taking any supplements that would enhance muscle-building. The following outcomes were assessed:

Muscle Thickness: Ultrasound imaging was used to obtain measurements of muscle thickness (MT). The reliability and validity of ultrasound in determining hypertrophic measures is reported to be very high (correlation coefficients of 0.998 and 0.999, respectively) when compared to the “gold standard” magnetic resonance imaging (Reeves, Maganaris & Narici, 2004). Moreover, ultrasound has a remarkable safety record with no known harmful effects associated with its proper use in adults (Nelson et al., 2009). Testing was carried out using a B-mode ultrasound imaging unit (Sonoscape S8 Expert; All Imaging Systems, Irvine, CA, USA). The technician, who was not blinded, applied a water-soluble transmission gel (Aquasonic 100 Ultrasound Transmission gel; Parker Laboratories Inc., Fairfield, NJ, USA) to each measurement site and a 5 MHz ultrasound probe was placed perpendicular to the tissue interface without depressing the skin. When the quality of the image was deemed to be satisfactory, the technician saved the image to the hard drive and obtained MT dimensions by measuring the distance from the subcutaneous adipose tissue-muscle interface to the muscle-bone interface as detailed in previous research (Schoenfeld et al., 2015a; Schoenfeld et al., 2015b). Measurements were taken on the right side of the body at four sites: biceps brachii, triceps brachii, medial quadriceps femoris, and lateral quadriceps femoris. For the anterior and posterior upper arm, measurements were taken 60% distal between the lateral epicondyle of the humerus and the acromion process of the scapula; for the quadriceps femoris, measurements were taken 50% between the lateral condyle of the femur and greater trochanter for both the medial (rectus femoris) and lateral (vastus lateralis) aspects of the thigh. Ultrasound has been validated as a good predictor of muscle volume in these muscles (Miyatani et al., 2004; Walton, Roberts & Whitehouse, 1997) and has been used in numerous studies to evaluate hypertrophic changes (Abe et al., 2000; Hakkinen et al., 1998; Nogueira et al., 2009; Young et al., 1983; Ogasawara et al., 2012). In an effort to help ensure that swelling in the muscles from training did not obscure results, images were obtained 48–72 h before commencement of the study and after the final training session. This is consistent with research showing that acute increases in muscle thickness return to baseline within 48 h following a resistance training session (Ogasawara et al., 2012). To further ensure accuracy of measurements, at least two images were obtained for each site. If measurements were within 10% of one another the figures were averaged to obtain a final value. If measurements were more than 10% of one another, a third image was obtained and the closest of the measures were then averaged.

Body Composition: Measures of body composition were determined by dual x-ray absorptiometry (DXA) imaging. Lean mass (total fat-free mass), fat mass, and percent body fat was assessed using a Hologic™ Discovery dual energy x-ray absorptiometer (DXA; Bedford, MA, USA). Subjects were instructed to refrain from exercise for 48 h and fast for 12-hours prior to each testing session. Upon arrival, participants had their height recorded using a SECA 242 instrument (242, SECA, Hanover, MD, USA) and weight recorded using TANITA electronic scale (Model TBF-310, TANITA, Arlington Heights, IL, USA). Prior to testing, all participants were instructed to remove any traces of metal that were present (cellphone, keys, jewelry, etc.). Participants then laid supine position dressed in either shorts or a gown, and were aligned on the table by a trained research assistant. Once a centered alignment was achieved, the participants were then instructed to lay still for approximately 7 min while a low dose of radiation scanned their entire body. For DXA measurements, previous test–retest reliability in our lab are as follows: Fat Mass: ICC = 0.998; Lean Mass: ICC = 1.00; percent body fat: ICC = 0.998. All DXA scans were conducted by the same technician, analyzed with the image compare mode for serial exam software feature, and followed strict manufacturer guidelines for calibration and testing procedures as per previously published work (Wilborn et al., 2013).

Maximal Strength: Upper and lower body strength was assessed by 1RM testing of the bench press (1RMBP) exercises followed by the parallel back squat (1RMBS). Subjects reported to the lab having refrained from any exercise other than activities of daily living for at least 48 h prior to baseline testing and at least 48 h prior to testing at the conclusion of the study. Repetition maximum testing was consistent with recognized guidelines as established by the National Strength and Conditioning Association (Baechle & Earle, 2008). In brief, subjects performed a general warm-up prior to testing consisting of light cardiovascular exercise lasting approximately 5–10 min. A specific warm-up set of the given exercise of five repetitions was performed at ∼50% of the subject’s estimated 1RM followed by one to two sets of 2–3 repetitions at a load corresponding to ∼60–80% of estimated 1RM. Subjects then performed sets of 1 repetition of increasing weight for 1RM determination. Three to 5 min rest was provided between each successive attempt. All 1RM determinations were made within five attempts. Subjects were required to reach parallel in the 1RMBS, defined as the point at which the femur is parallel to the floor, for the attempt to be considered successful as determined by the trainer. Successful 1RMBP was achieved if the subject displayed a five-point body contact position (head, upper back and buttocks firmly on the bench with both feet flat on the floor) and executed a full lock-out. 1RMBS testing was conducted prior to 1RMBP with a 5 min rest period separating tests. All strength testing took place using free weights. Recording of foot and hand placement was made during baseline 1RM testing and then used for post-study performance. All testing sessions were supervised by two fitness professionals to achieve a consensus for success on each attempt.

Statistical analysis

Data were analyzed using a linear mixed model for repeated measures, estimated by a restricted maximum likelihood algorithm. Treatment was included as the between-subject factor, time was included as the repeated within-subjects factor, time × treatment was included as the interaction, and subject was included as a random effect. Repeated covariance structures were specified as either Hyunh-Feldt or compound symmetry, depending on which structure resulted in the best model fit as determined by Hurvich and Tsai’s Akaike’s information corrected criterion (Hurvich & Tsai, 1989). As only significant main effects of time were observed, post-hoc analyses on main effects for time were done using multiple t-tests, with adjusted p-values from the simulated distribution of the maximum or maximum absolute value of a multivariate t random vector (Edwards & Berry, 1987). Effect sizes were calculated as the mean pre-post change divided by the pooled pretest standard deviation (Morris, 2008). Cohen’s D classification of small (0.2), medium (0.5), and large (0.8) were used to denote the magnitude of effects (Cohen, 1988). All analyses were performed using SAS Version 9.2 (Cary, NC, USA). Effects were considered significant at P ≤ 0.05. Data are reported as x ¯±SD unless otherwise specified.

Results

The total number of subjects initially enrolled was 59. During the course of the study, 38 subjects dropped out for the following reasons: Eight failed to follow up; 11 failed to comply with the study requirements; 10 did not have time in schedule to participate; four sustained an injury that disabled them from completing the testing protocol; three passed the deadline for study completion so their participation was suspended; and two moved away and thus were unavailable for testing sessions. Thus, 21 subjects ultimately completed the study. Attendance for those completing the study was 97.3%. All results are presented in  Table 1.

Table 1 Study outcomes.

	PRE T1	PRE T2	PRE T3	POST T1	POST T2	POST T3	P value for group	P value for time	P value for group by time interaction	PRE effect size T1–T3	POST effect size T1–T3	
Body weight (kg)	86.3 ± 17.8	85.4 ± 15.5	84.7 ± 15.9	80.3 ± 9.3	79.4 ± 9.1	79.6 ± 8.4	0.31	0.07	0.65	−0.12	−0.05	
BM (DEXA) (kg)	79.9 ± 17.3	79.1 ± 14.8	78.4 ± 15.3	74.1 ± 9.0	73.0 ± 8.8	73.4 ± 8.1	0.32	0.09	0.52	−0.11	−0.05	
Left arm TM (kg)	5.3 ± 1.0	5.2 ± 0.9	5.2 ± 1.2	4.6 ± 0.6	4.6 ± 0.6	4.6 ± 0.5	0.08	0.57	0.97	−0.08	−0.05	
Right arm TM (kg)	5.4 ± 1.0	5.3 ± 0.7	5.4 ± 1.0	5.0 ± 0.7	4.8 ± 0.6	5.1 ± 0.7	0.20	0.18	0.53	−0.01	0.10	
Left leg TM (kg)	14.4 ± 3.4	14.3 ± 2.8	14.1 ± 3.1	13.4 ± 1.9	13.3 ± 1.9	13.2 ± 1.8	0.39	0.45	0.96	−0.08	−0.08	
Right leg TM (kg)	14.8 ± 3.5	14.9 ± 3.1	14.7 ± 3.3	13.7 ± 2.0	13.8 ± 2.1	13.6 ± 1.9	0.34	0.67	0.93	−0.01	−0.06	
Total FM (DEXA) (kg)	12.2 ± 9.0	11.8 ± 9.3	10.9 ± 7.9	8.9 ± 3.5	8.1 ± 2.8	7.9 ± 2.4	0.24	0.001*	0.58	−0.20	−0.16	
BF% (DEXA)	14.1 ± 6.4	13.8 ± 7.4	12.9 ± 5.9	12.0 ± 4.5	11.1 ± 3.7	10.8 ± 3.2	0.34	0.002*	0.66	−0.23	−0.24	
Left arm FM (kg)	0.6 ± 0.3	0.5 ± 0.3	0.5 ± 0.4	0.5 ± 0.2	0.4 ± 0.1	0.4 ± 0.1	0.26	0.008*	0.80	−0.15	−0.23	
Right arm FM (kg)	0.5 ± 0.3	0.5 ± 0.3	0.5 ± 0.3	0.5 ± 0.2	0.4 ± 0.1	0.4 ± 0.1	0.25	0.09	0.52	−0.16	−0.19	
Left leg FM (kg)	2.4 ± 1.8	2.2 ± 1.6	2.1 ± 1.5	1.6 ± 0.7	1.5 ± 0.5	1.4 ± 0.5	0.17	0.0005*	0.42	−0.23	−0.12	
Right leg FM (kg)	2.5 ± 1.8	2.4 ± 1.8	2.3 ± 1.7	1.7 ± 0.6	1.6 ± 0.6	1.6 ± 0.4	0.16	0.02*	0.85	−0.15	−0.11	
Total LM (DEXA) (kg)	64.5 ± 8.9	64.6 ± 5.5	64.8 ± 7.4	62.6 ± 8.3	65.1 ± 12.2	63.0 ± 7.4	0.76	0.58	0.58	0.04	0.05	
Left arm LM (kg)	4.5 ± 0.7	4.4 ± 0.7	4.5 ± 0.8	4.0 ± 0.6	3.9 ± 0.6	4.0 ± 0.5	0.09	0.74	0.93	−0.05	0.02	
Right arm LM (kg)	4.6 ± 0.8	4.6 ± 0.5	4.6 ± 0.7	4.3 ± 0.6	4.2 ± 0.6	4.4 ± 0.6	0.25	0.09	0.55	0.03	0.19	
Left leg LM (kg)	11.3 ± 1.5	11.5 ± 1.1	11.4 ± 1.4	11.2 ± 1.8	11.2 ± 1.9	11.2 ± 1.7	0.78	0.91	0.81	0.04	−0.04	
Right leg LM (kg)	11.7 ± 1.6	11.9 ± 1.3	11.8 ± 1.5	11.4 ± 1.9	11.6 ± 1.9	11.4 ± 1.7	0.67	0.57	0.84	0.09	−0.02	
Biceps T	41.5 ± 4.9	40.9 ± 6.0	42.1 ± 6.3	36.3 ± 4.1	36.7 ± 4.1	39.2 ± 5.9	0.06	0.09	0.48	0.12	0.57	
Triceps T	51.5 ± 9.3	50.8 ± 9.3	51.9 ± 8.8	53.5 ± 7.5	50.2 ± 8.9	54.0 ± 6.5	0.74	0.23	0.61	0.05	0.06	
Lateral quad T	56.6 ± 4.7	55.0 ± 5.2	54.1 ± 4.7	54.9 ± 7.2	56.0 ± 7.3	53.5 ± 6.1	0.76	0.19	0.69	−0.40	−0.23	
Medial quad T	65.4 ± 6.7	66.9 ± 8.1	64.5 ± 11.8	67.6 ± 7.6	67.9 ± 8.1	68.6 ± 7.0	0.47	0.77	0.52	−0.13	0.14	
Squat 1-RM	159 ± 22	164 ± 23	165 ± 23	146 ± 28	150 ± 25	154 ± 21	0.23	0.003*	0.73	0.24	0.30	
Bench 1-RM	124 ± 16	126 ± 20	126 ± 18	117 ± 23	118 ± 23	121 ± 22	0.48	0.07	0.50	0.15	0.20	
Notes.

* Indicates statistically significant result.

T1 Baseline

T2 Midpoint

T3 Endpoint

Nutrition

Figures 1 and 2 graphically illustrate the energy and macronutrient intake of the subjects, respectively. There was no significant group by time interaction (P = 0.18) or group effect (P = 0.30) for self-reported calorie intake. There was a significant effect of time (P = 0.02), with calorie intake at T2 being significantly lower than T1 (adjusted P = 0.02). There were no significant interactions or main effects for self-reported protein or carbohydrate intake (P = 0.22–0.78). For self-reported fat intake, there was no significant group by time interaction (P = 0.43) or group effect (P = 0.35), but there was a significant effect of time (P = 0.0008), with fat intake being significantly lower at T2 and T3 compared to T1 (adjusted P = 0.001–0.02).

Figure 1 Self-reported kcal intake in pre-exercise supplementation (PRE-SUPP) and post-exercise supplementation (POST-SUPP) groups.

T1, Baseline; T2, Midpoint; T3, Endpoint. Data are presented as means ± SD. ∗, significantly different from T1 (P < 0.05).

Figure 2 Self-reported macronutrient intake in pre-exercise supplementation (PRE-SUPP) and post-exercise supplementation (POST-SUPP) groups.

T1, Baseline; T2, Midpoint; T3, Endpoint. Data are presented as means ±SD. ∗,  significantly different from T1 (P < 0.05).

Body mass

For body weight and DXA-determined total mass, probability approached significance for an effect of time (P = 0.07–0.09), with a tendency for weight and DXA-determined total mass to decrease from baseline to week 10 in both groups. For left-arm total mass, probability approached significance for an effect of group (P = 0.08), with group PRE-SUPP having a tendency for greater left arm total mass compared to group POST-SUPP. There were no other significant effects or interactions for body mass or segmental total mass. Effect sizes were small for both groups.

Fat mass

There was a significant effect of time for left arm fat mass (P = 0.008). Post-hoc analysis revealed significantly lower left arm fat mass at T2 and T3 compared to T1 (adjusted P = 0.01–0.02). Probability approached significance for right arm fat mass to decrease from baseline to week 10 (P = 0.09). For left leg fat mass, there was a significant effect of time (P = 0.0005). Post-hoc analysis revealed significantly lower left leg fat mass at T2 and T3 compared to T1 (adjusted P = 0.0004–0.01). Right leg fat mass also showed a significant effect of time (P = 0.02), with right leg fat mass being lower at T3 compared to T1 (adjusted P = 0.02). For overall fat mass, there was a significant effect of time (P = 0.001), with fat mass at T3 being significantly lower than T1 (adjusted P = 0.0004). Total DXA-determined body fat percentage showed a significant effect of time (P = 0.002), with T3 being significantly lower than T1 (adjusted P = 0.001). Effect sizes were small for both groups. Overall the findings show a modest reduction in body fat for both groups over the course of the study.

Lean mass

For left arm lean mass, probability approached significance for an effect of group (P = 0.09), with group PRE-SUPP having a tendency for greater left arm lean mass compared to group POST-SUPP. For right arm lean mass, probability approached significance for an effect of time (P = 0.09), with a tendency for right arm lean mass to increase from baseline to week 10. There were no other significant effects or interactions for total lean mass or segmental lean mass. Effect sizes were small for both groups. Overall the findings show little change in lean mass across groups.

Muscle thickness

For biceps thickness, probability approached significance for an effect of group (P = 0.06), with group POST-SUPP tending to be greater than group PRE-SUPP. In addition, probability approached significance for an effect of time (P = 0.09), with a tendency for biceps thickness to increase from baseline to week 10. There were no other significant effects or interactions for measures of muscle thickness. Effect sizes were small for both groups, with the exception of biceps thickness, which showed a moderate effect size in POST-SUPP. Overall the findings show a modest advantage for POST-SUPP on increases in biceps thickness, with minimal changes in other hypertrophic measures. Individual changes in muscle thickness are displayed in Figs. 3–5.

Figure 3 Biceps thickness.

Individual changes in biceps thickness for PRE and POST. Values in mms. T1, Baseline; T3, Endpoint.

Figure 4 Medial quadriceps thickness.

Individual changes in medial quadriceps thickness for PRE and POST. Values in mms. T1, Baseline; T3, Endpoint.

Figure 5 Lateral quadriceps thickness.

Individual changes in lateral quadriceps thickness for PRE and POST. Values in mms. T1, Baseline; T3, Endpoint.

Maximal strength

There was a significant effect of time for 1RM squat (P = 0.003), with T3 being significantly greater than T1 (adjusted P = 0.002). For 1RM bench, probability approached significance for an effect of time (P = 0.07), with a tendency for an increase from baseline to week 10. Effect sizes were small for both groups.

Discussion

To the authors’ knowledge, this is the first study to directly investigate muscular adaptations when consuming protein either immediately before or after resistance exercise in a cohort of trained young men. The primary and novel finding of this study was that, consistent with the research hypothesis, the timing of protein consumption had no significant effect on any of the measures studied over a 10-week period. Given that the PRE-SUPP group did not consume protein for at least 3 h post-workout, these findings refute the contention that a narrow post-exercise anabolic window of opportunity exists to maximize the muscular response and instead lends support to the theory that the interval for protein intake may be as wide as several hours or perhaps more after a training bout depending on when the pre-workout meal was consumed.

Both PRE-SUPP and POST-SUPP groups significantly increased maximal squat strength by 3.7% and 4.9%, respectively. Moreover, probability approached significance for greater changes in maximal bench press strength for PRE-SUPP and POST-SUPP, with increases of 2.4% and 3.3%, respectively. There were no significant differences in either of these measures between groups. Our findings are consistent with those of Candow et al. (2006), who found that consumption of a 0.3 g/kg protein dose either before or after resistance training produced similar increases in 1RM leg press and bench press in a cohort of untrained elderly men over 12 weeks. Conversely, the findings are somewhat in contrast with those of Esmarck et al. (2001), who found that consuming an oral liquid protein dose immediately after exercise produced markedly greater absolute increases in dynamic strength compared to delaying consumption for 2 h post-workout (46% versus 36%, respectively), although the values did not reach statistical significance. The reasons for discrepancies between studies is not clear at this time.

Neither group demonstrated significant gains in lean mass of the arms or legs over the course of the study. With respect to direct measures of muscle growth, probability approached significance for an increase in biceps brachii thickness (p = 0.06) while no significant changes were noted in the triceps brachii and quadriceps femoris. No interactions were found between groups for any of these outcomes. Results are again consistent with those of Candow et al. (2006), who found similar increases in muscle thickness of the extremities regardless of whether protein was consumed before or after training. Alternatively, our findings are in sharp contrast to those of Esmarck et al. (2001), who reported a 6.3% increase in muscle cross sectional area in a cohort of elderly men who received protein immediately after resistance training while those delaying consumption for 2 h displayed no hypertrophic changes. The findings of Esmarck et al. (2001) are curious given that numerous studies show marked hypertrophy in an elderly population where no specific dietary restrictions were provided (Frontera et al., 1988; Tracy et al., 1999; Ivey et al., 2000; Roth et al., 2001); it therefore seems illogical that delaying protein consumption for just 2 h post-exercise would completely eliminate any increases in muscle protein accretion. Moreover, subjects in Esmarck et al. (2001) study who consumed protein immediately post-workout experienced gains similar to that shown in other research studies that did not provide a timed protein dose (Verdijk et al., 2009; Frontera et al., 1988; Godard, Williamson & Trappe, 2002). Thus, there did not appear to be a potentiating effect of post-exercise supplementation in Esmarck et al. (2001) study. Considering the very small sample size of the non-timed group (n = 6), this calls into question the validity of results and raises the possibility that findings were due to a statistical anomaly.

Acute studies attempting to determine an “anabolic window” relative to the resistance training bout have failed to yield consistent results. In a similar way that temporal comparisons of nutrient administration in the post-exercise period have been equivocal (Levenhagen et al., 2001; Rasmussen et al., 2000), comparisons of whether protein/amino acid administration is more effective pre- or post-exercise have also been conflicting. Tipton et al. (2001) reported that 6 g essential amino acids (EAA) co-ingested with 35 g sucrose immediately pre-exercise resulted in a significantly greater and more sustained MPS response compared to immediate post-exercise ingestion of the same treatment. A subsequent investigation by Tipton et al. (2007) reported no difference in net muscle protein balance between 20 g whey protein ingested immediately pre- versus immediately post-exercise. Although it is tempting to assume that there is an inherent difference in whole protein versus free amino acids, Fujita et al. (2009) reported similar increases in post-exercise MPS when healthy, young subjects consumed a solution of EAA (0.35 g/kg/FFM)−1 and carbohydrate (0.5 g/kg/FFM)−1 versus being fasted prior to a bout of high-intensity lower body resistance training. Collectively, the acute data do not indicate conclusive evidence of a specific temporal dosing bracket where intact protein or amino acid administration enhances resistance training adaptations.

A caveat to our findings is that despite extensive counseling efforts to ensure that subjects maintained a consistent caloric surplus, both groups substantially reduced their energy intake from baseline. The reason for this discrepancy is not clear, but it can be speculated that subjects may have considered the supervised study an opportunity to lose body fat while gaining muscle, and thus taken it upon themselves to adjust energy intake accordingly. The reduction in calories over the study period resulted in a significant reduction in body fat, with losses of 1.3 and 1.0 kg for PRE-SUPP and POST-SUPP, respectively. It is well-documented that maintaining a caloric deficit is suboptimal for building muscle. In the absence of regimented exercise, there is generally a loss of lean body mass; for every pound of weight lost, approximately 25% comes from FFM (Varady, 2011). Adoption of a higher protein diet and regular resistance training can attenuate these losses and even promote slight increases in muscle mass depending on factors including training status, initial body fat levels, and the extent of caloric restriction (Garthe et al., 2011; Stiegler & Cunliffe, 2006). That said, to achieve robust hypertrophic gains requires a sustained non-negative energy balance (Garthe et al., 2013). Taken in this context, our findings indicate that PRE-SUPP and POST-SUPP strategies are similarly effective in enhancing muscle development during calorically-restricted fat loss and cannot necessarily be extrapolated to a mass-building program that incorporates an energy surplus.

The study had several notable limitations. First, the sample size was fairly small, increasing the possibility of null findings due to type II errors. Second, subjects trained using a 3-day-a-week resistance training program. Given that subjects were resistance-trained men with ample lifting experience, it is possible that a higher volume routine might have produced different results. Third, the free-living nature of the study prevented close monitoring of activity levels outside of the research setting, and it remains possible that this may have impacted results. Fourth, the study did not have a wash-out period; thus, differences between the study protocol from the subject’s pre-training routine may have influenced results from a novelty standpoint. Fifth, we did not monitor energy expenditure outside of training sessions as well as during sleep; it is unclear whether the timing would have affected such outcomes. Sixth, self-report dietary records are known to have a high degree of variance from actual nutritional intake (Mertz et al., 1991); thus, caution must be used in the interpretation of food-consumption data. Seventh, the study employed a 3 day-per-week total body RT routine. Although this routine has been shown to produce significant hypertrophic increases in the target population (Schoenfeld et al., 2015a; Schoenfeld et al., 2015b; Schoenfeld et al., 2016), it remains possible that results may have differed if subjects trained with a split-body routine that allowed for a greater total weekly training. Eighth, DXA has been shown to be prone to potential confounding by changes in hydration status (Nana et al., 2012). Although we attempted to minimize these changes by instructing subjects to refrain from physical activity and food consumption prior to testing, it remains possible that body composition changes were influenced by alterations in hydration. Finally, muscle thickness was measured only at the middle portion of the muscle. Although this region is generally considered to be indicative of whole muscle growth, we cannot rule out the possibility that greater changes in proximal or distal muscle thickness occurred in one protocol versus the other.

Conclusion

It has been hypothesized that protein ingestion in the immediate post-exercise period is the most critical nutrient timing strategy for stimulating MPS, and on a chronic basis, optimizing muscular adaptations. In the face of this common presumption, the comparison of protein timed immediately pre- versus post-exercise has both theoretical and practical importance due to individual variations in the availability and/or convenience of protein dosing relative to training. In the present study, the presence of a narrow “anabolic window of opportunity” was not demonstrated as reflected by the fact that PRE-SUPP group showed similar changes in body composition and strength to those who consumed protein immediately post-exercise. Across the range of measures, there were no meaningful results consistently attributable to pre- versus post-exercise protein ingestion. The implications of these findings are that the trainee is free to choose, based on individual factors (i.e., preference, tolerance, convenience, and availability), whether to consume protein immediately pre- or post-exercise.

Nevertheless, the conditions of the present study warrant consideration. Despite specific instruction to maintain a caloric surplus, subjects fell into hypocaloric balance (objectively indicated by bodyweight and fat mass reductions). This raises the possibility that the results might be limited to scenarios where there is a sustained energy deficit. Previous work recommends covering the bases by ingesting protein at 0.4–0.5 g/kg of lean body mass in both the pre- and post-exercise periods (Aragon & Schoenfeld, 2013). This seems to be a prudent approach in the face of uncertainty regarding the optimization of nutrient timing factors for the objectives of muscle hypertrophy and strength.

Supplemental Information

File S1 Data file for all individual outcomes

Click here for additional data file.

We would like to extend our gratitude to Dymatize Nutrition for providing the protein supplements used in this study.

Additional Information and Declarations

Competing Interests

Author Contributions

Human Ethics

Data Availability

James Krieger is an employee of Weightology.

Brad Jon Schoenfeld conceived and designed the experiments, wrote the paper, prepared figures and/or tables, reviewed drafts of the paper.

Alan Aragon wrote the paper, reviewed drafts of the paper.

Colin Wilborn performed the experiments, contributed reagents/materials/analysis tools, wrote the paper, reviewed drafts of the paper.

Stacie L. Urbina and Sara E. Hayward performed the experiments, wrote the paper, reviewed drafts of the paper.

James Krieger wrote the paper, prepared figures and/or tables, reviewed drafts of the paper.

The following information was supplied relating to ethical approvals (i.e., approving body and any reference numbers):

University of Mary Hardin-Baylor Institutional Review Board.

The following information was supplied regarding data availability:

The raw data has been supplied as Supplementary File.

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
