# Peer review of "Pre- versus post-exercise protein intake has similar effects on muscular adaptations"

_PeerJ, doi:10.7717/peerj.2825_

## Round 0.1 · original submission · Major Revisions

The three reviewers are overall quite impressed with your manuscript but have highlighted a number of areas that you need to address before it can be considered for publication.

·

Basic reporting

The article is very well written and adherent to all PeerJ policies. The introduction flows well; it succinctly but thoroughly narrates the reader through prior research in this area, identifies confounding limitations, and the resultant need for this collection of methods to contribute to our understanding of the area. The only slight interruption to introductory flow is the inclusion of the prior review (lines 103-108) in the area at a late stage after already detailing some specific original research. It is clearly a relevant reference but would probably fit better earlier in the introduction as a prelude to ensuing text.

The reference list contains multiple minor errors and inconsistencies such as use of capitalisations and journal abbreviations; errors I am sure will be an easy remedy once due attention is paid.

There is no table title.

Experimental design

There are no issues with experimental design aspects. Methods are very well detailed.

Validity of the findings

There is no issue with validity of the findings, and all conclusions are based, as much as reasonably possible given stated limitations, on results. The major caveat of the caloric deficit issue is well considered given it clearly confounds our ability to address the stated hypothesis. I would expect the initial power analysis did not allow for the observed dropout rate, and with the trained cohort the smallest worthwhile change in key variables would not have been great enough to warrant typical analyses with the eventual low N. Given that there was a much higher non-completion rate than what the authors were probably anticipating based on prior experience and typically encountered training study issues, I suggest an addition to the statistical modelling utilised, and reliance on the interpretive value of p values, would be to present the individual changes of some key variables of interest such as Total LM and others at the authors' discretion. There are 253 data entries in Table 1, and although it provides needed detail, it's a fairly hard read. Perhaps graphical representation of change % scores within and between groups would provide the reader with a more expedient option to review results generally. Then, where there are specific aspects of interest the table and/or raw data could be further scrutinised. For example, some spaghetti or scatter plots such as those suggested by Weissgerber et al., (2015) PLoS Biol 13(4): e1002128. doi:10.1371/journal. pbio.1002128. or a forest plot of mean change and confidence intervals would allow efficient presentation of multiple representative variables.

·

Basic reporting

The manuscript is well-written and presents an interesting area of research related to resistance training.

Experimental design

- The choice of training intervention may not have been appropriate for the subjects based on their previous experience.
- While an attempt was made to control for diet, activity outside of the exercise intervention was not monitored.
- I am not sure why there was a mid-point for testing? This needs to be clarified.
- Maybe regions of interest (ROI) analyses for the DXA should be performed rather than for the whole legs and arms e.g. upper arm, upper leg? This will help provide more site specific changes, if there is any?

Validity of the findings

Although the results suggest there is not a narrow post-exercise anabolic window to maximize muscular response, the intervention overall did not provide an adequate stimulus for subjects to increase muscular hypertrophy. The results are difficult to decipher as it seems that the characteristics of the subjects (> 1 years resistance training experience) were not compatible with the resistance training intervention used. Furthermore, dietary compliance was reduced towards the end of the intervention and I also suspect changes in physical activity outside of the training intervention.

Additional comments

Line 153: flat barbell “bench” press

Line 155: dumbbell “bicep” curl

Lines 161-164: It is confusing why a 10RM testing was performed if the prescribed intensity was based on 1RM % (i.e. 75% 1RM).

Lines 171-173: Why were loads adjusted rather than have the subjects perform forced repetitions (i.e. with a spotter). Reducing the load means that the intensities being used are altered which may have an effect on muscular strength gains.

Lines 186-188: What was the compliance? 100%? If it was, I strongly doubt the level of information was that thorough. This will need to be covered in the Discussion.

Line 206: to “the” hard drive

Line 226 – 235: Since body composition was assessed by only the DXA it is very important that the protocol is described. It is well known that hydration status and non-fasted conditions can lead to large errors in measurement when scanning participants. This should be addressed.

Line 236-237: 1RM testing ‘in’ the …. Should be ‘of’ the

Line 190-191: “Testing was conducted prior to the initial training session (T1), at the mid-point of the study (T2), and after the final training session (T3)”. Please explain how the mid-point testing did not interfere with the training intervention?

Line 288: No need to include Blood Pressure and Heart Rate results as they are not relevant.

Lines 292-302: This could be summarized more clearly. What do the findings mean? The same with the Lean Mass and Muscle Thickness sections.

Line 370: Add reference.


Line 394: This is a major issue with this study.


Other Comments:

You need to include the training volume (repetitions x load) for both groups at the three time points to help the reader understand the rate of progression.

Was there any wash-out period from the subjects’ previous training? Also, the resistance training intervention may have been significantly different to the subjects’ previous training? This may have had an influence on the rate of training adaptations.

I am not sure why a whole body and not a split-routine were used for this study? Did the subjects usually follow one style of training? It is quite unusual for experienced resistance trainers to follow a training program that is recommended for novice trainers? This is possibly the reason why little change in muscular hypertrophy was found for both groups.

How much protein were the subjects consuming per day at the three time points? Were you aiming for ~ 2 grams/kg body weight? How about calories?

Why did you not record energy expenditure outside of training sessions as well as sleep (since they are university students). These are potentially factors that may have influenced your study’s results and should be discussed.

·

Basic reporting

No comments

Experimental design

No Comments

Validity of the findings

-While there were no significant differences between strength in the PRE vs. POST groups, the POST group was greater and the increases in strength themselves are less than 5% absolutely, even a small % difference between groups can approach significance. Further, with the high dropout rate in the study, if there are more subjects in the study, is it possible this could have approached significance?

Additional comments

-How long were the training sessions? The timecourse of whey protein to stimulate muscle protein synthesis is 2-3 hours and thus, while the supplement was consumed PRE workout, isn’t it possible that amino acid levels remained elevated POST workout and therefore still in the theorized ‘anabolic window?’ Did the authors measure plasma levels of amino acids in response to feeding?

-As the subjects were at least 1 year resistance trained, was there any reason that the programs were not periodized? Obviously, the subjects increased strength, however is it possible a periodized program would have enhanced these effects and possibly produced differences between groups?

-Do the authors care to speculate why total kcal intake went down. I acknowledge this is not the author’s fault, it would be useful to speculate on why their energy intake dropped.

---

## Round 0.2 · Minor Revisions

We applaud you for taking on board most of the initial concerns of the reviewers, however the reviewers still highlight a few small issues with you to address prior to this paper being accepted for publication.

·

Basic reporting

The spaghetti plots now included don't really serve the purpose as per my original suggestion. Although they present the individual data, they don't easily provide a visual representation of between group differences. The reference I quoted includes an excel template to expedite usage as needed, and it allows a nice clear presentation of the between group differences in the one figure to compliment the details in table and in results text. It also cuts down the sheer number of figures.

Experimental design

No further comments

Validity of the findings

No further comments

·

Basic reporting

Please address the dietary record compliance which was alluded to in my previous concerns (lines 186-188). Like I said previously it is very unlikely that subjects would be 100% compliant with reporting everything consumed or accurate with portion of the foods /drinks consumed (Macdiarmid et al., 1998). Please address this by reporting missing meals, days etc.

Macdiarmid, J and Blundell, J. Assessing dietary intake: Who, what and why of under-reporting. Nutr Res Rev 11: 231-253, 1998.

Experimental design

“We used ultrasound to measure site-specific changes, which is a more accurate method for determining subtle alterations in muscle mass over time.”
Thus the suggestion for DXA ROI changes was that you only assessed muscle thickness at one site on the muscle of interest and there is a strong possibility that changes in muscle thickness may not be evenly distributed along the length of a muscle (Wakahara et al., 2013). It is also possible that changes in muscle thickness may have differed in the transverse plane (medial-lateral), which has been previously suggested (Wakahara et al., 2015). This is also a limitation of this study.

Wakahara, T, Fukutani, A, Kawakami, Y, and Yanai, T. Nonuniform muscle hypertrophy: its relation to muscle activation in training session. Med Sci Sports Exerc 45: 2158-2165, 2013.

Wakahara, T, Ema, R, Miyamoto, N, and Kawakami, Y. Increase in vastus lateralis aponeurosis width induced by resistance training: implications for a hypertrophic model of pennate muscle. Eur J Appl Physiol 115: 309-316, 2015.

For the DXA, hydration status? Was this considered? (Nana et al., 2012)

Nana, A, Slater, GJ, Hopkins, WG, and Burke, LM. Effects of daily activities on dual-energy X-ray absorptiometry measurements of body composition in active people. Med Sci Sports Exerc 44: 180-189, 2012

Validity of the findings

Totally agree about the importance of resistance training frequency however a split routine would allow for a greater resistance training volume over a given week. Your previous study did show greater muscle thickness (MT) in forearm flexors with no other statistical differences between other MT measures or 1RM (Schoenfeld et al., 2015). In addition, while your previous study used trained subjects, the resistance training performances at baseline were greater for subjects in this study compared to your previous one (1RM BP = ~120 vs. <100 kg; 1RM SQ = ~150 vs. <120 kg, respectively). Heterogeneity? There are too many other factors to consider (training variables) to dismiss the potential benefit of a split compared to total body routine. I do think something needs to be mentioned about total body vs. split routine training because of the frequent use of the latter training strategy among individuals targeting muscle hypertrophy.

Schoenfeld BJ, Ratamess NA, Peterson MD, Contreras B, Tiryaki-Sonmez G.. J Strength Cond Res. 2015 Jul;29(7):1821-9.).

---

## Round 0.3 · accepted · Accept

We wish to thank the authors for addressing the minor comments posed by the two reviewers. The paper is now worthy of publication in PeerJ. Please note the very minor amendment to the spaghetti plot requested by reviewer one.

·

Basic reporting

Now fine but the spag plot y values don't need to be to .000.

Experimental design

No further comments

Validity of the findings

No further comments

·

Basic reporting

No comments.

Experimental design

No comments.

Validity of the findings

Ni comments.